# Plain language summaries: A systematic review of theory, guidelines and empirical research

**Marlene Stoll**[1,2]*, **Martin Kerwer**[1], **Klaus Lieb**[2], **Anita Chasiotis**[1]

**1** Leibniz Institute for Psychology (ZPID), Trier, Germany, **2** Leibniz Institute for Resilience Research (LIR), Mainz, Germany

* ms@leibniz-psychology.org

**Data Availability Statement:** All relevant data are within the paper and its Supporting Information files.

**Funding:** The authors received no specific funding for this work.

## Abstract

Plain language summaries (PLSs) have been introduced to communicate research in an understandable way to a nonexpert audience. Guidelines for writing PLSs have been developed and empirical research on PLSs has been conducted, but terminology and research approaches in this comparatively young field vary considerably. This prompted us to review the current state of the art of the theoretical and empirical literature on PLSs. The two main objectives of this review were to develop a conceptual framework for PLS theory, and to synthesize empirical evidence on PLS criteria. We began by searching Web of Science, PubMed, PsycInfo and PSYNDEX (last search 07/2021). In our review, we included empirical investigations of PLSs, reports on PLS development, PLS guidelines, and theoretical articles referring to PLSs. A conceptual framework was developed through content analysis. Empirical studies investigating effects of PLS criteria on defined outcomes were narratively synthesized. We identified 7,714 records, of which 90 articles met the inclusion criteria. All articles were used to develop a conceptual framework for PLSs which comprises 12 categories: six of PLS aims and six of PLS characteristics. Thirty-three articles empirically investigated effects of PLSs on several outcomes, but study designs were too heterogeneous to identify definite criteria for high-quality PLSs. Few studies identified effects of various criteria on accessibility, understanding, knowledge, communication of research, and empowerment. We did not find empirical evidence to support most of the criteria we identified in the PLS writing guidelines. We conclude that although considerable work on establishing and investigating PLSs is available, empirical evidence on criteria for high-quality PLSs remains scarce. The conceptual framework developed in this review may provide a valuable starting point for future guideline developers and PLS researchers.

## Introduction

Good research practices include the publication and dissemination of results as well as their honest and transparent communication [1, 2]. It is further argued that the public—or at least relevant stakeholders—should have access to research, not only technically but also

**Competing interests:** The authors have declared that no competing interests exist.

intellectually [3]. This means that the public needs to be able to understand what the researchers have done, what the results mean and which practical implications can be drawn from them [3]. This information is mostly communicated by researchers via scientific publications, which is—among other factors—further motivated by funding agencies as well as promotion and tenure committees that expect researchers to publish in high impact scientific journals [4, 5]. The target audience of such publications and relevant gatekeepers, namely the editors and reviewers of high impact journals, are other researchers. Consequently, this traditional way in which scientific publications are written and published requires researchers to stress the scientific implications of their research. The traditional communication of scientific findings therefore constitutes a scientific 'bubble' in which scientists communicate with each other about the meaning of their findings [6]. As is typical for such group formation processes, this bubble results in its own special type of language, shared knowledge as well as implicit and explicit norms, which makes scientific communication harder to understand for those outside the scientific bubble [7]. Such a context provides a breeding ground for the evolution of idiosyncratic professional jargon. This lack of plain, easily comprehensible language hinders the public from directly accessing scientific articles [8].

One viable and ready solution that accounts for the information needs of the public (or of gatekeepers such as journalists) within the current research ecosystem are plain language summaries (PLSs)—a lay-friendly summary format of scientific research [8]. Each PLS is thought to provide a brief overview of a study and its main practical implications in a manner that is understandable to laypersons. There are various institutions that currently provide some form of lay-friendly summary formats of scientific studies. Especially institutions in biomedicine have been very prolific in this regard [9]. Yet, there is no consensus on PLS criteria (e.g., what defines a PLS and what defines a high-quality PLS). Guidance on how to write a PLS is scattered and often relates to specific purposes of the respective institutions that provide the PLS [9]. To evaluate whether a PLS is effective in fulfilling its aim (e.g., to be understandable for laypersons), empirical research focusing on PLSs as a subject has been intensified in recent years [e.g., 10–15].

To shed light on the status quo of PLS writing guidelines and the empirical research on PLSs, we present a systematic review of the established writing guidelines with their respective criteria and of the empirical evidence on the effectiveness of PLSs in terms of defined outcomes (e.g., understandability).

## Theoretical background

Although the use of PLSs has been steadily growing, there appears to be no broad consensus on the terminology underlying these lay-friendly summaries [9, 16]. In the present paper, we consistently refer to 'plain language summaries' (PLSs), irrespective of their designation in the referenced publications. PLSs, in this review, are defined as relatively brief textual summaries of scientific publications—targeted at laypersons—which complement the respective traditional abstracts and summarize the whole scientific article in a balanced, easily understandable manner.

The considerable variation in the terms that are used to refer to PLSs [9] may be partly related to the different *aims* that the specific parties (e.g., authors, stakeholders or publishers) pursue with their PLSs. On the one hand, some are guided by the explicit aim of translating medical evidence into a PLS to promote understanding thus enabling patients to actively engage in the process of shared decision making [17] or to increase patient engagement in clinical research [18, 19]. Terms used for these PLSs include, for example, 'clinical trial results summaries for laypersons' [20, 21], 'patient lay summaries' [22] or 'consumer summaries' [23].

On the other hand, publishers of scientific journals may primarily aim to increase the impact and accessibility of the published articles [18]. They might use terms such as 'lay summaries' [e.g., 24–26], 'translational abstracts' [27] or 'lay abstracts' [e.g., 24]. An overview of the variety of PLS terminology can be found in Shailes [16] for PLSs of scientific research in general, and in FitzGibbon et al. [9] for biomedical PLSs.

Besides common aims (e.g., increasing the accessibility of research), PLS formats differ in various *characteristics*. Considerable variation exists even for basic formal characteristics, such as the length of the PLS or the language they are written in [28]. For example, the *Journal of Applied Sport Psychology* presets a word limit of 50 words for their 'lay summaries' [29] which are expected to be written in English language, whereas Cochrane specifies a limit of up to 700 words and invests considerable efforts in translating the PLSs into multiple languages [30, 31]. Similar to the differences in the terminology used to label the PLSs, these differences in text characteristics may be related to the specific aims that are pursued by the specific parties for providing PLSs. If, for example, the aim is to reach out to other researchers to foster interdisciplinary exchange, providing English PLSs may be a reasonable choice. Otherwise, if the aim is to reach out to the broader public in a more general way, PLSs additionally have to be written in languages other than English, to maximize accessibility. Such differences might not only manifest themselves in the varying languages or word limits of PLSs, but also in the varying recommendations of style or content in PLS writing guidelines (that, e.g., outline what the PLS should explain, or whether additional background information or statistical information should be provided). PLSs may thus differ considerably in their characteristics, depending on the specific guidelines on which they are based (if at all) and which *criteria* these guidelines specify. Criteria in this context are standards or restrictions of characteristics (e.g., specific word limits or approaches for dealing with technical terms).

In empirical studies on the effectiveness of PLSs, such criteria are systematically varied and analyzed with regard to certain defined *outcomes* that are supposed to reflect the aims in terms of measurable, operationalized variables. For example, researchers could investigate the PLS aim of increasing public empowerment by evaluating whether people who read a PLS with plain language explanations of technical terms perform better in a knowledge test about the contents of the summary than people who read a PLS without such plain language explanations.

In sum, theory on the concept of PLSs refers to the interconnectedness of these four main subject areas: PLSs serve specific (1) *aims* (e.g., to improve laypersons' understanding of scientific findings) that are accomplished by certain (2) *characteristics* (e.g., linguistic attributes) with their related target values or (3) *criteria* (e.g., avoiding technical terms). Criteria can also be subject to empirical investigations (e.g., comparing PLSs with different criteria) of certain (4) *outcomes* (e.g., knowledge tests). We illustrate the specific meaning of these terms within this article with the following analogy: Assume the topic is not PLS but a cake you want to bake for guests. A cake's aim or purpose could be to "taste delicious". Characteristics of a cake are, among others, its ingredients ("flour", "sugar"). Criteria are target values of the characteristics that aim to fulfill its purpose ("3 cups flour", "2 cups sugar"). An outcome to test whether the "aim" of the cake is achieved is to ask your guests if the cake tasted delicious.

In this review, PLSs are considered both a research subject as well as a service or intervention to make research understandable for laypersons. For this systematic review on PLS research and theory, we will therefore not only synthesize research on the subject using an *ontological* approach by addressing the question of what the topic constitutes and what it distinguishes from other topics (this refers to PLS characteristics). Since PLSs are a specific form of 'research translation' service or intervention, our review additionally collects and discusses information provided by the extant literature on the purpose of a PLS (a *finalistic* approach

referring to the aims of a PLS), and specifically focusing on what must be undertaken to fulfill this purpose (a *normative* approach referring to the criteria of a PLS) and what steps can be taken to evaluate its effectiveness (a *measurement-related* approach referring to the outcomes of empirical studies on PLSs). Thereby, a comprehensive overview of four main subject areas, namely the characteristics and aims of PLSs as well as PLS criteria and measured outcomes, is provided. We particularly focus on existing guidelines and empirical investigations of criteria with regard to defined outcomes, to identify evidence-based criteria for writing PLSs.

## Objectives

The overarching aim of this review is to give a comprehensive overview of the current understanding of PLSs in the scientific literature and of evidence for their effectiveness. On that account, we will systematize theoretical research and empirical evidence on PLSs by considering theoretical articles (including literature reviews) and opinion pieces, empirical studies as well as writing guidelines. With that said, the aim of our review is twofold.

First, we intend to establish a conceptual framework of PLSs by outlining finalistic, ontological, normative and measurement-related approaches to capturing this research field. For this purpose, we will synthesize the systematically reviewed literature with regard to four questions relating to the main subject areas of this review:

1. What are the *aims* of a PLS (finalistic approach; i.e., what purpose do PLSs serve)?

2. What are the *characteristics* of a PLS (ontological approach; i.e., what constitutes PLSs)?

3. Which *criteria* define a PLS or are considered to constitute a high-quality PLS (normative approach; i.e., what exactly should PLSs be like)? Which criteria have been empirically investigated in the context of PLSs?

4. Which *outcomes* of PLSs have been investigated (measurement-related approach; i.e., how are PLSs evaluated)?

Second, we will sum up and integrate the empirical evidence on PLSs against the background of this research topic's main subject areas: the aims, characteristics and criteria of PLSs as well as outcomes in PLS research. By doing so, we will be able to identify current gaps in the empirical research on PLSs and reflect on the achievement thus far in providing evidence-based guidelines for writing PLSs.

## Methods

### Eligibility criteria

For this review, we searched for scientific publications on PLSs (i.e., empirical and theoretical articles) as well as for published PLS guidelines. The term 'PLSs', in this review, refers to research summaries that accompany a scientific publication with the aim to translate published scientific evidence from language that is geared toward expert audiences to language that is geared toward lay audiences. Examples for such scientific publications are original research reports, meta-analytical studies or clinical study reports. This definition of a PLS explicitly does not include popularized science news articles, blog posts or patient-education materials because these are not direct translations of scientific publications but (research) outputs or publications in their own right. In our interpretation, apart from translating the evidence described in research articles, the only autonomous scientific contribution a PLS may make is to report the evaluation of risk of bias or provide contextual information to the reader (e.g., by highlighting the practical implications of the evidence or providing additional background

information). This criterion is based on the consideration that evaluating risk of bias is just another way of translating the quality of a scientific finding for a non-expert audience. This audience is expected to have no experience with scientific standards, and additional information on practical implications may help them understand the scientific finding. Finally, with 'PLSs' we only refer to textual approaches for translating evidence and, therefore, not to infographics, videos or podcasts. Accordingly, we searched for publications and guidelines that investigate, discuss or describe PLSs. We stipulated three criteria of inclusion for the investigated, discussed or described PLSs:

A. The PLS is a summary of published scientific evidence (i.e., from a primary study or a systematic review).

B. The PLS aims at a lay readership.

C. The PLS uses the same communication format as the original scientific publication evidence (i.e., text).

We included English and German records of the following study types:

- quantitative and qualitative studies or reviews of such studies in which

    a. characteristics, outcomes, criteria or aims of PLSs are investigated, or

    b. PLS criteria or guidelines that combine several criteria are developed or evaluated;

- guidelines on how to write PLSs;

- theoretical articles (e.g., opinion pieces, theoretical discussions, reviews, editorials, comments) that clearly focus on PLS characteristics, outcomes or aims, or on PLS writing criteria or guidelines.

### Information sources and information search

We systematically searched Web of Science, PubMed, PsycInfo and PSYNDEX (last search on July 2, 2021) using the search terms specified in S1 File. After the first selection process (see below), we performed a backward reference search of included articles. Additionally, we searched the websites of journals which publish PLSs for journal-specific PLS writing guidelines, and searched the web for more such guidelines.

### Study selection

In the first step, titles and, if necessary, abstracts were screened to exclude irrelevant records based on the above-mentioned inclusion criteria. In a second step, the remaining potentially relevant records were assessed for eligibility by a full text screening. In both steps, double screenings by two independent researchers were performed (Fig 1). Discrepancies were discussed, and, if unsolvable, the decision was made by a third, independent rater.

### Data collection process and data items

Three of the authors (MS, MK and AC) independently extracted information about the four main subject areas (aims, characteristics, criteria and outcomes) in the form of text passages from the selected reports. As our aim was to give a descriptive overview of theoretical and empirical research on PLSs, we did not perform an additional quality assessment of the reports. Each report was evaluated by one of the authors, and the respective informative text passages were compiled in tables.

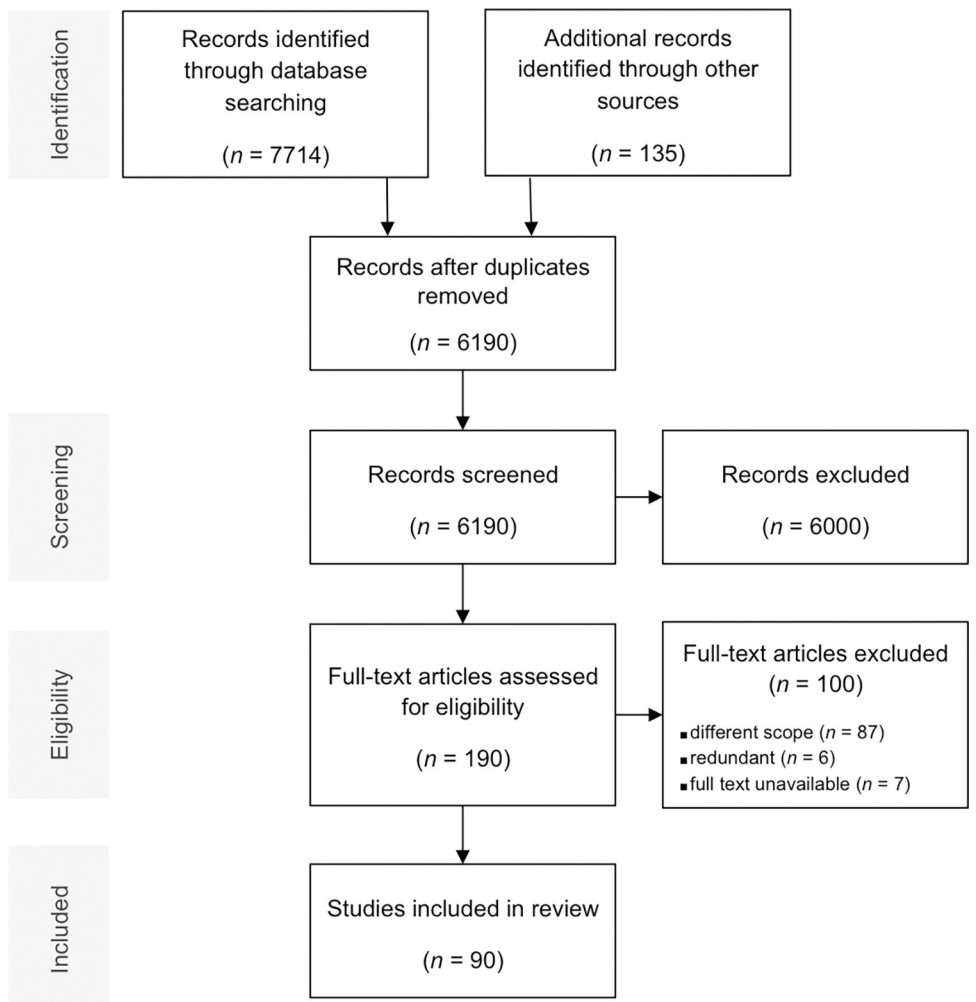

**Fig 1. Study selection flow diagram.**

After the text passages had been recorded, MS went through all passages again and collated them to the respective original record to ensure proper rendition. Study procedures were not preregistered.

## Analysis

We based our approach for analyzing and summarizing the information from the full texts on qualitative content analysis [32]. As described above, we first extracted information in the form of text passages separately for each of the four subject areas (aims, characteristics, criteria, outcomes) that corresponded to the four approaches of describing the PLS research field from theory. In the next step, we categorized the information that we obtained for each subject area by means of identifying and labeling homogeneous groups of information. Three of the authors (MS, MK, AC) with expertise on PLSs each independently worked through the extracted text passages and proposed categories for the four subject areas. Discussing these categories, it became clear that the subject areas 'aims' and 'outcomes' as well as the subject areas 'characteristics' and 'criteria' share similar categories. Subsequently, we discussed how the contents of these subject areas may be linked to each other. The result of this process was a first

draft version of our conceptual framework. We discovered that outcomes constitute operationalizations of aims and were therefore subordinated to the aims categories. Similarly, criteria appeared to be operationalizations and specifications of characteristics. In the next step, the three experienced authors proposed categories for the subject areas 'aims' and 'characteristics' only. The proposed categories and their supporting rationales were discussed by the three authors until an agreement for a set of categories for 'aims' as well as 'characteristics' was obtained. We then mapped the information on outcomes to the aim categories with the goal of determining the extent to which the empirically investigated outcomes reflect the theoretical aim categories. We also mapped the information on criteria to the characteristic categories of PLSs to determine the specifications of different categories of PLS characteristics (i.e., criteria) that have been found to or are suggested to distinguish a PLS from other text formats or to constitute a high-quality PLS.

The information was mapped by the first author (MS) who allocated each text passage on PLS outcomes to one of the PLS aims categories and each text passage on PLS criteria to one of the PLS characteristics categories, respectively. Afterwards, two other authors (MK, AC) reviewed the categorization and proposed changes in case of disagreement. These cases were discussed until consensus was reached. During and after this process, the final conceptual framework was developed. Based on this framework, we performed a narrative review of the empirical evidence on PLSs and compiled a comprehensive table of criteria reported in PLS guidelines.

## Results

### Study selection

We identified 7,714 records through database searching and 135 records through other sources (see above). After title and full text screening (Fig 1), we included 90 studies in our review.

### Study types

Of the 90 included records, 36 (40%) were theoretical and 33 (37%) were empirical articles (Fig 2). Twenty-one articles (23%) were guidance-related articles. Of the 36 theoretical articles, 7 (19%) were reviews and 29 (81%) were opinion pieces, editorials or comparable articles. Of the 33 empirical articles, 15 (45%) described experiments that quantitatively compared different formats of PLSs ($n = 6$) or compared PLSs with other summary formats ($n = 9$), 3 (9%) were studies that qualitatively compared different formats of PLSs ($n = 2$) or compared PLSs with other summary formats ($n = 1$), and 15 (45%) were studies that evaluated or investigated one specific type of PLSs. Of the 21 guidance-related records, 17 (81%) were guidelines and 4 (19%) were other studies related to PLS guidance (e.g., guideline development). All studies are listed in S1 Table.

Information on PLS aims, characteristics, criteria and outcomes was extracted from all 90 studies. Specific information on criteria for writing a PLS was extracted in detail from the 17 guidelines. Specific information on empirical evidence of PLSs was additionally extracted and summarized in detail from the eight empirical studies that quantitatively or qualitatively compared different forms of PLSs.

### Aims of PLSs

PLS aims can be divided into six categories which we labeled 'Accessibility', 'Understanding', 'Knowledge', 'Empowerment', 'Communication of Research' and 'Improvement of Research' (see below for the description of each category).

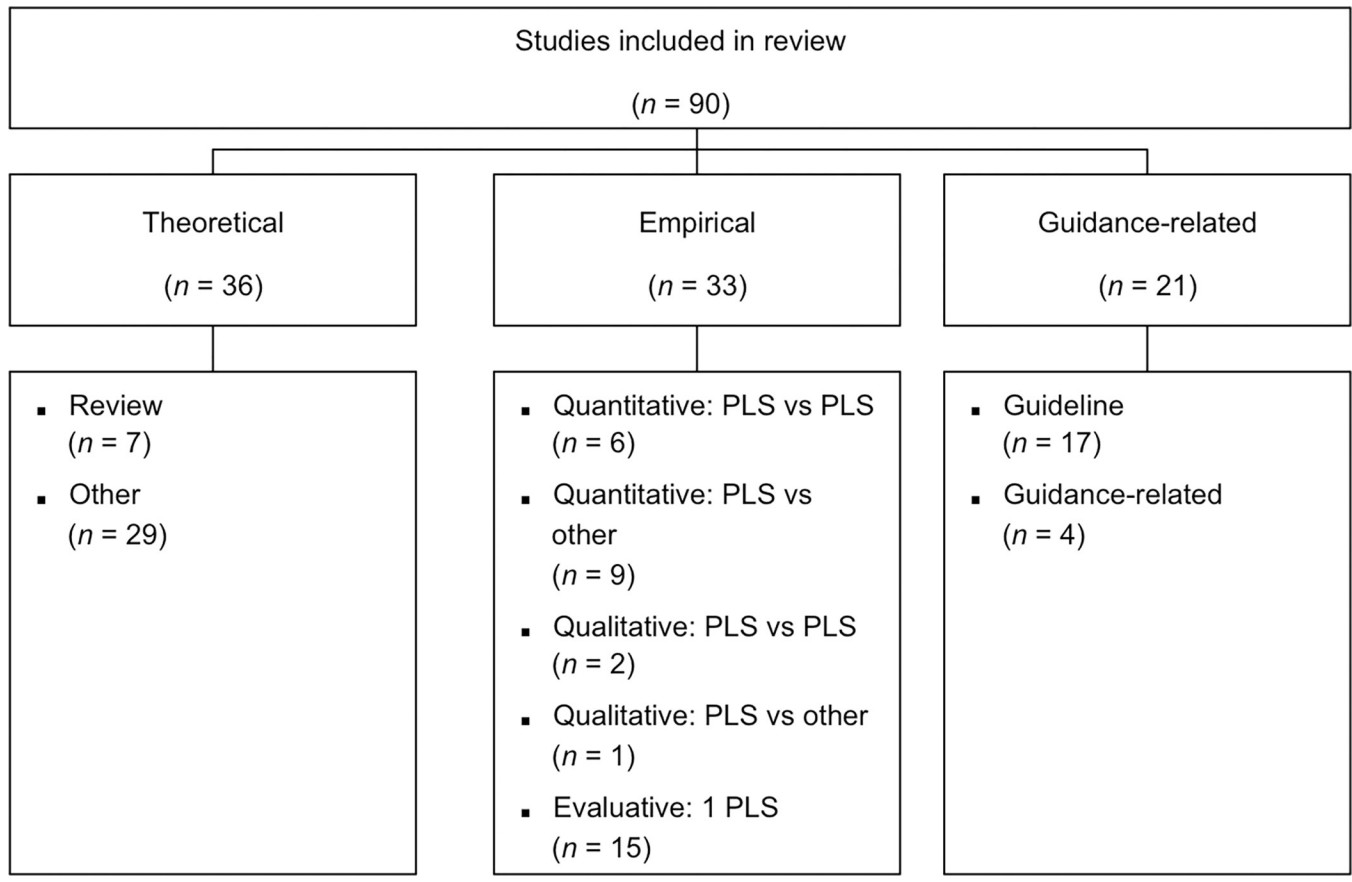

**Fig 2. Study types of studies included in this review.**

The aims category 'Accessibility' comprises PLS aims that are geared towards providing laypersons with low-threshold information on research that is easy to find, highly visible, freely accessible, attractive, and appealing to non-experts in general or to specific non-experts such as in teacher training [33]. This category also includes information on the technical accessibility of the PLS [34].

- The aims category 'Understanding' comprises PLS aims that are geared towards providing laypersons with information on research (including research questions, methods and results) that is understandable.

- The aims category 'Knowledge' comprises PLS aims that are geared towards increasing laypersons' knowledge about specific subjects based on scientific evidence.

- The aims category 'Empowerment' comprises PLS aims that are geared towards enabling laypersons to make informed, self-determined decisions, and to foster public participation in decision processes.

- The aims category 'Communication of Research' comprises PLS aims that are geared towards enhancing the communication and dissemination of research by addressing a broad audience. Thereby, the trust in and impact of science on daily decisions as well as on political decisions and actions is thought to increase. For example, Phung et al. [35] stresses the

interaction of different audiences, and Wada et al. [36] name the PLS as a tool to communicate with project funders.

- The aims category 'Improvement of Research' comprises PLS aims that are geared towards making a contribution to the improvement of research practice itself, for example through increased transparency, exactness and improvement of writing style as well as through higher engagement in discussions on the relevance of research. Furthermore, PLSs are thought to facilitate interdisciplinary communication. They may improve research by strengthening public support for the research enterprise [37], enhancing transparency [38], and by providing the opportunity for science to engage in the media ecosystem [26].

It must be noted that the first four aim categories are dependent on each other, however they present distinct aims: To empower laypersons to make informed decisions based on scientific findings (Empowerment category), laypersons need to have evidence-based knowledge (Knowledge category), which in turn is only possible if they understand the information about the evidence (Understanding category). For that, the core requirement is that the information is accessible (Accessibility category). Aims of the Communication of Research category represent ideal aims, often pictured as a bridge over the gap between academia and the public, while aims of the Improvement of Research category include effects that these actions have on research itself. These effects are measurable irrespective of how PLSs are (subjectively) received.

To determine to what extent the empirically investigated outcomes reflect the theoretical aims categories, we mapped text passages on PLS outcomes to the theoretical aims categories (see Table 1 for examples of the matching process). In the next section, we narratively describe all outcomes, structured by the aims categories these outcomes can be subordinated to.

## Outcomes of PLSs

Many outcomes could be linked to aims of the Accessibility category. In their empirical studies, researchers investigated the readability of PLSs [6, 13, 15, 21, 22, 28, 43–48], participants'

**Table 1. Aims categories with matched exemplary text passages of aims and outcomes.**

| Aims categories | Aims in the literature (examples) | Outcomes in the literature (examples) |
|---|---|---|
| Accessibility | "A key aspect of improving access to knowledge is to ensure not only that the content of the resource is appropriate but also that the format in which it is presented is fit for purpose." [14, p.2] | "Usability and accessibility were framed as positive questions and were measured on a seven-point Likert scale: 1 (strongly disagree) to 7 (strongly agree)." [12, p.186] |
| Understanding | "PLS help to make scientific research understandable. . . by describing complex research using nontechnical language that can be easily understood" [9, p.2] | Item "Understanding": "I understand this research more after reading this abstract. (0) Not at all. . . (4) Very Much" [39, p.6] |
| Knowledge | ". . . evidence summaries are. . . instrumental resources for translating research to inform knowledge" [40, p.93] | "Comprehension of the content of the summary format was assessed by a brief knowledge test with four multiple choice questions for each PLS" [10, p.3] |
| Empowerment | "Making it more likely that the findings of the research will be used to make a difference to service users' lives." [25, p.2] | "Subsequent sections included items on. . . satisfaction with the way the information prepared them for decision making" [23, p.5] |
| Communication of Research | "The PLS is considered a main building block for dissemination of the review to the end-users of health information." [41, p.3] | Item: "On which platforms have you shared or reused an eLife digest?" [42, Fig 3] |
| Improvement of Research | "For researchers to commit the time and effort to learn the skills and write good summaries, they need to believe that public engagement is one possible approach to improving the quality, relevance and impact of their work." [25, p.8] | "The primary aim of this study is to investigate the extent to which published reports of research into the effects of physiotherapy interventions provide plain-language summaries. The secondary aims are to determine: (i) if the proportion of these reports that include plain-language summaries is increasing over time; (ii) if the inclusion of a plain-language summary for a randomised trial is associated with trial quality" [43, p.355] |

enjoyment or preference of PLSs in general [10, 11, 39, 49], participants' satisfaction with text length [13, 22, 42] or participants' judgment regarding the usability of the PLS [3, 12, 23, 28, 50, 51]. To measure these outcomes, researchers, for example, computed readability indices, such as the Flesch reading ease score [13, 44, 43, 47, 48], or participants were asked whether information was easy to find in the PLS [50].

We further identified a wide range of outcomes that were linked to aims of the Understanding category and Knowledge category. Aims of the Understanding category were investigated by asking participants whether they perceived the text as understandable and by asking them about their user experience [21–23, 28, 38, 39, 42, 45, 49, 50, 52]. A typical outcome asked readers to report how easy or difficult it was for them to read the PLS [e.g., 13], which was measured on a Likert scale. We further observed outcomes that covered objective knowledge gain (Knowledge category) [10–12, 14, 21, 38, 39, 46, 49, 50, 52, 53]. To assess aims of the Knowledge category, researchers typically used multiple choice formats (e.g., "This research focuses on: a) HIV, b) FIV, c) Influenza, d) I don't know" [39, p. 6]) or open-ended questions that referred to the content of the PLS ("What is external cephalic version in breech position—how would you describe that term to a friend?" [49]). Yet other outcomes included whether participants were able to name the purpose of the summary [12], how they judged the effectiveness of a treatment that was described in the PLS [10] or their judgment on the quality of evidence [12].

Even more specific were outcomes that we linked to aims of the Empowerment category [10, 12, 13, 21, 23, 28, 38, 42, 50]. Readers were asked, for example, whether they would use PLSs to make certain decisions (mostly investigated in the context of health care, [e.g., 50]) or how supported and prepared they felt to make certain decisions [e.g., 23, 38] or to evaluate or discuss the subject with others [13, 38] after reading the PLS.

In the empirical studies included here, aims of the Communication of Research category were only examined indirectly [3, 12, 13, 21, 28, 39, 42, 45, 50, 53]. Readers were asked, for example, who they think the summary has been written for [21], how important or relevant they perceived the related study or research to be [45] or whether they shared or reused the PLS [42].

Even less empirical research on PLS included outcomes that related to the Improvement of Research category [43, 45, 50, 52]. One study surveyed PLS volunteer contributors and asked them whether writing PLSs improved their skills in writing, editing and time management, and whether it increased their confidence in lay audience communication, enhanced their understanding of the specific scientific literature, or helped in their development as a scientist [52]. This category was also represented in a study that asked PLS readers about the perceived usefulness of science for the public after having read the PLS [45]. One study examined the association between the inclusion of a PLS and trial quality [43].

## Characteristics of PLSs

The typical PLS characteristics can be divided into six categories. We have labeled these 'Linguistic Attributes', 'Formal Attributes', 'General Content', 'Presentation of Results', 'Presentation of Quality of Evidence' and 'Contextual Attributes' and we provide a description of each category in the following.

'Linguistic Attributes' are PLS characteristics that encompass the tone or style of the language, the choice of words (e.g., handling of jargon and technical terms), or the text difficulty.

- 'Formal Attributes' are PLS characteristics on the formal level, such as word limits, standardized formulations, prespecified headlines, or inclusion of graphs or tables. This also includes whether a PLS follows a formal structure, for example, characterized by the use of headlines and subheadlines or paragraphs.

- 'General Content' comprises PLS characteristics that concern a PLS' content (e.g., whether background information or key messages are included), and the alignment of contents (e.g., a prespecified alignment of introduction, description of methods, results and conclusions).

- 'Presentation of Results' are PLS characteristics that encompass the presentation of results in a PLS (e.g., whether an effect size is mentioned, or the way statistical terms are handled).

- 'Presentation of Quality of Evidence' comprises PLS characteristics that include the presentation of the quality of evidence in a PLS, for example whether GRADE-system (Grading of Recommendations, Assessment, Development and Evaluation) results or authors' conflicts of interest are reported.

- 'Contextual Attributes' are PLS characteristics that concern the general context of the PLSs, for example the specific process of drafting, production or publication. These include information on the review process, on technical accessibility and the target group.

To learn more about what constitutes a high-quality PLS, we searched the literature für PLS criteria. As these criteria can be considered specifications of PLS characteristics, we matched text passages on PLS criteria to the characteristics categories (Table 2). In the following section, we describe all criteria we found in the PLS literature, structured according to the characteristics categories they can be subordinated to.

## Criteria of PLSs

We were able to glean the most information on criteria in published guidelines or other guidance-related publications. A few empirical studies investigating the criteria were also available. Criteria were also specified in opinion pieces albeit rarely. In the following, we summarize the criteria that have been mentioned or investigated in the PLS literature and their link to PLS characteristics categories.

Table 2. Characteristics categories with matched exemplary text passages of characteristics and criteria.

| Characteristics categories | Characteristics in the literature (examples) | Criteria in the literature (examples) |
|---|---|---|
| Linguistic Attributes | "Plain language summaries. . . are thus diverse in style, word usage, and possibly in literacy requirements." [44, p.2] | "Avoid complex or meaningless terms and phrases: Many terms used in academic English are either overcomplicated or contain no useful information. Examples include terms such as 'virtually' or 'literally' or archaic language (e.g. amidst, whilst), as well as verb choices such as 'purchase' used in place of the simpler 'buy'." [25, p.4] |
| Formal Attributes | "Plain language summaries. . . have different word counts depending on the journal." [39, p.2] | "Recommended length of a PLS: The target length is 600 to 750 words." [54, p.5] |
| General Content | "The content of lay summaries has evolved over the past 5 years, with earlier versions not always including all the elements now required." [55, p.264] | "The text should provide answers to the essential questions: Who, What, Where, When, Why, How? For example, the reader should easily be able to find answers to questions such as 'By whom was the research funded, and why?'" [25, p.5] |
| Presentation of Results | "For example, in current plain language summaries authors use a variety of words to express. . . the magnitude of the effect of the interventions." [17, p.495] | "Cochrane's Plain Language Expectations for Authors of Cochrane Summaries (PLEACS) standards recommend that it is not essential to provide numerical information in PLSs, but if there are numbers presented, the presentation should be consistent, comprehensive to the lay population in terms of absolute effects, and framed as natural frequencies" [10, p.2] |
| Presentation of Quality of Evidence | "However, research is still lacking for other aspects of how to present research findings. For instance, we know little about. . . how to convey the quality of this evidence." [56, p.567] | "Recommendation #7: Indicate level of evidence supporting risk estimates (eg, gold and silver)" [23, p.5] |
| Contextual Attributes | "Approaches. . . include. . . paying specific attention to the PLS as part of the editorial process, and/or moving the responsibility of writing the PLS to dedicated writers." [57, p.2] | "We recommend that:. . . PLS are developed by PLS authors, although support may be sought (eg from journal editorial staff and/or patient organizations) to ensure appropriate readability" [9, p.6] |

Most of the criteria referred to characteristics of the Linguistic Attributes category [3, 6, 8, 13, 15, 18, 20–25, 27, 40, 43–45, 50, 52, 54, 55, 58–78]. For example, in some articles, it was recommended to use active rather than passive voice [8, 15, 25, 69, 73, 75], and some articles provide more or less specific recommendations regarding the use of jargon: avoid jargon [54, 64, 71–73, 75–78]; use only 2% jargon in the whole text [6]; use only short words or sentences, avoid polysyllabic words, acronyms or abbreviations [18, 20, 50, 60, 73, 75, 77, 78]; avoid technical terms [66, 72]; avoid potentially misunderstood words [20] or define terms if necessary [18, 22, 66, 73, 75, 77]. Using technical terms and defining them may be reasonable if oversimplifying terms or concepts leads to inappropriate, misleading or inaccurate content [78].

Further, we matched substantial criteria to the characteristics of the Formal Attributes category [15–17, 20–24, 27, 29, 38, 42, 45, 50, 54, 55, 58, 61, 62, 64, 67–69, 71, 73, 78–84]. These included, for example, the exact word limit per PLS, ranging from 50 words [29] to 750 words [54], recommendations on structuring the text or the use of headings [21, 73, 78], or whether visual images should be used [20, 23, 50, 84].

Other frequently specified criteria referred to characteristics of the General Content category [15, 20–23, 25, 27–29, 38, 42, 50, 56–58, 62–69, 71–79, 81, 82, 84–87]. These included recommendations on how to formulate the first sentence or paragraph of the PLS: for example, the first sentence should summarize the purpose of the (clinical) trial [87]; the first sentence should make clear to the reader who the summary has been written for and why it has been written [21]; the first sentence should include something that most readers can relate to [66]. Furthermore, recommendations were made as to which information should be included in the PLS: for example, provide answers to the essential questions: Who, What, Where, When, Why, How? [25]; outline three main elements: primary scientific question, what was learned, and why it matters [60] or what should be avoided (e.g., PLS authors should avoid promotional content [20]).

We further identified a variety of criteria for characteristics that could be linked to the Presentation of Results category [3, 17, 20, 22, 23, 27, 45, 50, 54, 55, 57, 59, 61, 66–70, 75, 78, 83]. Some articles stated that statistical significance should be clearly explained if required, whereas *p*-values, confidence intervals or standard deviations should be avoided [e.g., 22]; others made recommendations to report sensitivity, specificity and prevalence in natural frequencies [83]; one article concluded from research with focus groups that numbers should be completely omitted [45]. Furthermore, one publication suggested presenting outcome probabilities in multiple ways and with consistent denominators [23]. A guideline by Cochrane included the rule that results of no more than seven outcomes should be reported in the PLS [e.g., 68].

Criteria that we were able to link to characteristics of the Presentation of Quality of Evidence category were only rarely mentioned in the investigated articles [23, 68, 69, 78]. Of the four articles we found, one specified the importance of reporting risk estimates in PLSs with the corresponding level of evidence [23], and another specified the necessity of publishing the researchers' conflict of interest statements alongside PLSs [78]. Further, the Cochrane guidelines specified that the overall quality of the evidence should be reported as well as any factors that might affect the confidence in the results (e.g., bias risks such as conflicts of interest [68]).

Lastly, we found criteria that referred to the characteristics of the Contextual Information category [8, 9, 15, 18, 20, 22–24, 27, 38, 42, 50, 52, 55, 57, 59, 63, 64, 67, 72, 73, 75, 78, 80, 84, 86–91]: For example, publications included specifications regarding PLS authorship [8, 9, 38, 64, 80, 87], and whether the submission of a PLS is or should be mandatory [18, 75, 89]. Another main contextual subject focused on the writing and publication process: There were specifications of where or how the PLS is or should be published (e.g., open access publishing [see 9, 80]) and some articles included detailed descriptions outlining the writing and

publishing process of the PLS [52, 73, 78, 84, 90]. Occasionally, recommendations were made to engage patients or members of the public in the PLS creation process [73, 84, 90, 91].

We found guidelines for writing PLSs from a variety of professional associations such as the American Psychological Association, APA [27], Cochrane [68, 69, 92] and the Campbell Collaboration [54], from funders of research [73], as well as from scientific journals [61, 71, 72, 75, 77]. We also identified guidelines that specifically refer to the EU regulation on summarizing the results of clinical trials [20, 65, 70] as well as general guidance in writing PLSs for scientific writers [24, 25, 66]. The criteria mentioned in all 17 guidelines are provided in S2 Table.

## Conceptual framework

The derived categories for aims and characteristics as well as their relations with outcomes and criteria are illustrated in Fig 3. This conceptual framework of our synthesis of the PLS literature depicts the four main subject areas of theory and research on PLSs. It indicates the main categories of a PLS's ascribed aims and characteristics together with examples of the respective outcomes (which are operationaliziations of aims) and criteria (which are specifications of characteristics). The finalistic approach, namely, to ask about the purpose of PLSs, determines its elements (ontological approach) and thereby, its characteristics. These characteristics are specified by certain criteria which determine what a PLS should be like (normative approach). The eligibility of such criteria is evaluated by analyzing their impact on certain outcomes in empirical studies, which, in turn, guide the development of adequate criteria. Outcomes constitute the operationalizations of aims in empirical studies (measurement-related approach), and thereby, the effect of a PLS on certain outcomes guides the decision concerning its suitability with regard to certain prespecified aims.

## Empirical evidence on PLSs

The included empirical articles on PLSs were highly heterogeneous with respect to study design, use of terminology and operationalization. Therefore, it was not possible to conduct a meta-analysis to quantitatively synthesize the study effects. Instead, the results of the 33 empirical articles on PLSs are narratively summarized, separated by study type. First, evaluative articles and articles in which one type of PLSs was compared to other summary formats are briefly summed up. Second, empirical articles that compare different forms of PLSs are described in more detail, matching the investigated criteria with our characteristic categories and the investigated outcomes with our aim categories.

## Empirical articles that evaluated one type of PLSs

We identified 15 empirical articles on PLSs that evaluated one type of PLSs, for example by investigating how readers reacted to a certain PLS, or how easy they can be found. Researchers evaluated Cochrane PLS that address common health issues [11, 57], summaries of studies from the Newcastle Cognitive Function after Stroke cohort [45], consumer summaries of Cochrane Musculoskeletal Reviews [23], lay summaries on how package sizes affect the consumption of food, alcohol and tobacco [28], lay summaries on open access journal articles [3], lay summaries of pragmatic pediatric clinical trials [19], lay summaries on HDBuzz, a knowledge translation website with a focus on research on Huntington's disease [52], PLSs of clinical trials in general [21, 93, 94], PLSs of clinical trials in Japan [95] and PLSs posted as 'eLife digests' [8, 66]. There was also one study that investigated how the commitment of volunteer PLS authors can be raised [96] and one study that developed and tested a PLS template in detail [78].

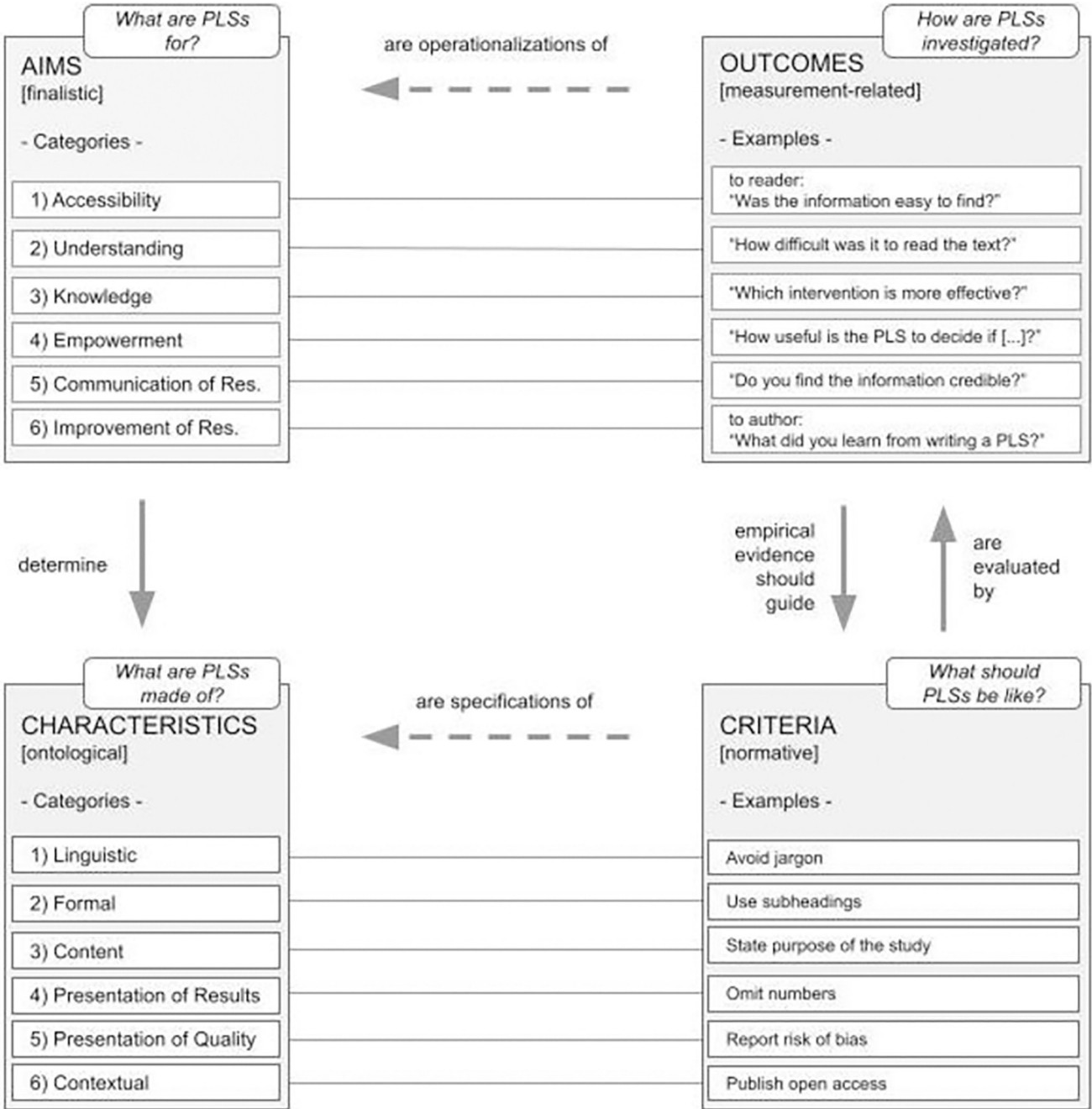

**Fig 3. Conceptual framework of aims, characteristics, criteria and outcomes investigated in PLS research.**

### Empirical articles that compared one type of PLSs to other summary formats

Ten other empirical articles investigated the format of a PLS in general by comparing participants' responses after having read a PLS to (the same or other) participants' responses after having read another type of scientific text summary. PLSs were compared to scientific abstracts [6, 44, 47, 49, 53], infographics [49] or graphical abstracts [39], blog posts [50, 97], blogshots

[11], podcasts [53], transcribed podcasts [53], Wikipedia articles [50], video abstracts [39], press releases [44], newspaper health articles [48] and systematic reviews with and without summary of findings tables [46]. There were statistically significant indications for the superiority of blogshots over both PLSs and Wikipedia articles in terms of ease of use, user preference and aesthetical judgment [50]; of videos and PLSs over graphical abstracts and scientific abstracts in terms of comprehension, a feeling of understanding and a feeling of enjoyment [39]; of PLSs over systematic reviews with and without summary of findings tables in terms of clarity and accessibility of information [46]; of infographics over PLSs in terms of reading experience and user-friendliness, but not of knowledge [49]; and of newspaper articles over PLSs in terms of readability [48]. Furthermore, scientific abstracts were found to be less readable [47] and to contain more jargon [6] than PLSs, but the amount of jargon in PLSs was higher than recommendations for public understanding stipulate [6].

## Empirical articles that compared different types of PLSs

We identified eight empirical articles that investigated specific criteria of PLSs by analyzing which criteria have an effect on defined outcomes. Two qualitative studies compared different PLS versions and asked readers about their experiences, and six quantitative studies investigated how different PLS versions affected readers' response patterns (e.g., in knowledge tests). We linked the criteria investigated in these studies to their respective characteristics categories and the outcomes investigated in these studies to the respective aims categories (Fig 3).

The results of two qualitative studies were reported in Ellen et al. [98] and Glenton et al. [56]. Ellen et al. [98] investigated the feedback of 18 Canadian health system managers and policy makers on PLSs of systematic reviews. In their study, they presented three different PLSs to each participant and conducted semi-structured interviews. They found that participants preferred structured text (bullet points, tables) with up-front key messages, including details on background, methods and applicability. Also, participants preferred evidence quality ratings (according to the GRADE method) in the summaries. Text blocks, unstructured texts and summaries with less background information were elements less preferred by the participants. Ellen et al. [98] also note that participants preferred a longer, structured PLS format over a shorter, unstructured format, suggesting that readers did not mind reading longer documents as long as they were well structured and could easily be scanned. Hence, the characteristics categories that were addressed in this study were Formal Attributes, General Content and Presentation of Quality of Evidence.

Glenton et al. [56] developed three different versions of a PLS and conducted semi-structured interviews with 34 members of the public in Norway, Argentina, Canada and Australia. Thereafter, a modified version of a PLS was developed and retested to produce a final version. The three preliminary versions of the PLS differed in their presentation of results: one version using qualitative statements only, a second version using both qualitative statements and numbers, and a third version using qualitative statements and additionally numbers and symbols in a table. The authors found that most participants preferred the third version. The results of this study showed participants' difficulty in interpreting effect sizes, confidence intervals and continuous outcomes. Challenges faced by participants included understanding the concepts of effect size, intervention effect and evidence quality as well as the difference between a systematic review and an individual study. The research group was able to partially improve understanding by adding symbols with explanations for evidence quality and by rephrasing the introductory text describing the concept of a systematic review. The characteristics categories that were analyzed in this study were thus Formal Attributes, General Content, Presentation of Results and Presentation of Quality of Evidence.

**Table 3. Quantitative studies comparing PLSs with PLSs: Criteria, outcomes and results.**

| Study (Sample Size) | Criteria | Outcome | Results |
|---|---|---|---|
| Santesso et al. (2015) [12] ($n = 143$) | new format vs. current format | knowledge test; comprehension test; usability survey; preference | new > current for all outcomes except comprehension test |
| Silvagnoli et al. (2020) [13] ($n = 167$) | readability level: low vs. medium vs. high | preference | medium > low; medium > high |
| Alderdice et al. (2016) [14] ($n = 813$) | conclusion vs. no conclusion; certain vs. uncertain findings | knowledge test | uncertain findings: conclusion > no conclusion certain findings: no significant effect |
| Buljan et al. (2020) [10], Trial 1 ($n = 91$) | positive framing vs. negative framing | knowledge test, usage of the described treatment; comprehension test | no differences |
| Buljan et al. (2020) [10], Trial 2 ($n = 245$) | natural frequencies vs. percentages | comprehension test; preference; knowledge test | no differences |
| Kirkpatrick et al. (2017) [15] ($n = 60$) | original PLS vs. PLS rewritten by author vs. PLS rewritten by independent writer | understanding survey; reading ease | both rewritten versions > original in reading ease; no further sign. effects |
| Kerwer et al. (2021) [38] ($n = 166$) | PLS with subheadings vs. PLS without subheadings | comprehensibility; knowledge acquisition; credibility; ability to evaluate the study; ability to make a decision | with subheadings > without subheadings for all outcomes |

Only experimental conditions that investigate PLS criteria against each other are listed; further comparisons (e.g., with other summary formats) are not reported here.

In the following section, we describe the six quantitative studies that systematically varied criteria of PLSs and compared them with regard to different defined outcomes. The experimental conditions, outcomes and results with the respective characteristics and aims categories are summarized in Table 3 (see S3 Table for a more detailed version).

Santesso et al. [12] compared a new PLS format for Cochrane Reviews to the format that was currently recommended at the time of their study. The new format was more structured and results were presented not only qualitatively, but also quantitatively. The criteria that were tested in their experiment were the differences between the new format and the current format. They cover the characteristics of the Formal Attributes, Presentation of Results and Presentation of Quality of Evidence categories. The investigated outcomes cover the aims of the Accessibility, Understanding and Knowledge categories: Participants who read the new format performed better in test of knowledge-related outcomes and, overall, judged the new format more accessible, despite demonstrating low comprehension of the purpose of the summary, with no statistical difference between the two formats [12].

Silvagnoli et al. [13] investigated how the complexity of PLSs affected the readers' preference for the text format, thereby testing an aim of the Accessibility category. The criteria that were tested in this experiment were three different complexity levels of PLSs in terms of readability scores (a Linguistic Attribute characteristic). The results showed that medium-level PLSs with a readability that corresponds to a reading age of 14–17 years were preferred most, compared to low-level and high-level complexity PLSs [13].

Alderdice et al. [14] investigated PLSs of Cochrane Reviews with uncertain findings and certain findings and compared a PLS format with a conclusion to a PLS format without a conclusion (General Content category). The outcome was a multiple choice test that can be linked to aims of the Knowledge category. They found a statistically significant difference only in PLSs with uncertain findings: Participants who read the uncertain PLS without a conclusion performed worse than participants who read the uncertain PLS with a conclusion [14].

Buljan et al. [10] conducted two trials that investigated the framing of numerical findings of Cochrane Review PLSs. In trial 1, they varied whether results in the PLS were positively framed or negatively framed and in trial 2, they varied whether trial effectiveness and side effects were

presented as natural frequencies or percentage scores. Thus, both trials investigated criteria that can be linked to characteristics of the Presentation of Results category. Outcomes in this study were operationalizations of aims of the Knowledge and Empowerment categories. There were no differences on these outcomes between the experimental groups in either trial [10].

Kirkpatrick et al. [15] compared two strategies to improve the quality of PLSs in the National Institute for Health Research (NIHR) Journals Library. They compared original PLSs to PLSs that were rewritten either by the author of the original scientific publication or by an independent professional writer, both with the help of a guideline. These experimental conditions represent characteristics of the Contextual Information category. The outcomes in this study were perceived ease of understanding (Understanding category) and reading ease (Accessibility category). The results revealed no statistically significant difference in terms of ease of understanding and that both rewritten versions were significantly easier to read than the original [15].

Kerwer et al. [38] compared PLSs with subheadings to PLSs without subheadings, varying a characteristic from the Formal Attributes category. The investigated outcomes in this study cover the aims of the categories Understanding, Knowledge, Empowerment, and Communication of Research. The authors found that participants rated PLSs with subheadings as more comprehensible and more credible than PLSs without subheadings. Subheadings compared to no subheadings were also related to higher knowledge acquisition and a higher perceived ability to evaluate the corresponding study and to make a decision based on the PLSs [38].

Summing up the empirical evidence on comparison of different PLSs, we note that an effect of specific criteria on outcomes was found in the following cases: PLSs are more accessible if they are written on a medium text level [13], and if they are rewritten with the help of a guideline by either the same or an independent writer [15]. PLSs impart more knowledge if they are presented with a conclusion than without a conclusion—but only if the study has uncertain findings [14]. PLSs in general impart more knowledge if they are presented in a more structured format [12, 38], and participants state that they prefer PLSs that are structured, use quality ratings and present sufficient background information [98] as well as PLSs that provide results with qualitative statements that are accompanied by numbers and symbols in tables [56]. PLSs with subheadings facilitated the communication of research as they were perceived to be more credible than PLSs without subheadings. Also, readers' empowerment—their perceived ability to make decisions based on the PLSs and to evaluate the veracity of the corresponding study—was higher in PLSs with subheadings compared to PLSs without subheadings [38].

No statistically significant effects of criteria were found in the following cases: There were no statistically significant differences in the perceived accessibility between PLSs that were rewritten by the scientific author versus rewritten by an independent writer [15] or in the ease of understanding between the original PLS and the rewritten version [15]. There were furthermore no statistically significant effects on understanding and knowledge outcomes between PLSs in a general new format versus a current format [12]; on knowledge outcomes between PLSs that had a conclusion and those with no conclusion in cases where findings of the study were certain [14]; between positive versus negative framing of results as well as between reporting results as percentages versus natural frequencies [10].

Interestingly, most of the studies we found investigated their research questions with samples that were not representative for the general public. Furthermore, sample sizes, overall, were comparably small: Four of the quantitative studies investigated samples between $n = 60$ and 245 [10, 12, 13, 15, 38]. One exception is Alderdice et al.'s [14] study with a sample of $n = 813$ [14]. In quantitative and qualitative studies, participants in these samples were often highly educated [13], students [10, 14, 38], or very selective, such as authors of research reports

[15] or health care managers and policy makers [98]. Only two studies used samples with members of the (general) public and patients that came from five different nations [12, 56].

## Empirical foundation of guideline criteria

Comparing the empirical evidence on the effectiveness of PLS criteria with criteria that are mentioned in guidelines (S2 and S3 Tables), it becomes clear that the empirically investigated criteria cover only a fraction of the entirety of criteria that are mentioned in the guidelines. The Cochrane guidelines on preparing PLSs of Cochrane intervention reviews [92] are grounded on empirical studies, two of which are also included in this review [12, 56]. Due to these studies' holistic approach—testing the whole format rather than single criteria—they provide a solid empirical basis for the Cochrane guidelines. However, this approach makes it difficult to deduce the effectiveness of single criteria and to generalize the results to other guidelines. The guideline of the Campbell Collaboration refers to the Cochrane guidelines and its empirical rationale [54], while the Summary of Clinical Trial Results for Laypersons-guideline is based on general health literacy principles [20]. Other guidelines that we found refer to this publication of the EU expert group and provide no further rationale [65, 70]. The guideline prepared by Duke [25] is presented as a synthesis of various other guidelines and advice for writing PLSs. However, in all other guidelines that we examined, we did not find any (empirical) rationale for the choice of criteria for writing a PLS. We conclude that current PLS guidelines only rarely provide any empirical rationale for their choice of criteria.

## Discussion

The aim of this review was twofold: First, we intended to develop a conceptual framework unifying the topic of PLSs in the scientific literature, and second, we aimed to synthesize empirical evidence on PLSs by integrating it into the conceptual framework.

To develop a comprehensive conceptual framework, we took into account different approaches that complement each other. Applying a finalistic approach, we scrutinized the aims of PLSs reported in the literature. We found that aims can be classified into the categories of Accessibility, Understanding, Knowledge, Empowerment, Communication of Research and Improvement of Research. This classification of aims resembles Nutbeam's [99] prominent classification of health literacy levels. This is particularly noteworthy as the majority of articles on PLSs that were included in our review derive from medical or public health journals. Thus, we must bear in mind that the current PLS literature has a strong focus on PLSs of medical research, with the aim to increase health literacy. According to Nutbeam, health literacy can be classified into three progressive ability levels: The first level, 'functional health literacy', refers to the ability to understand health information and to have sufficient knowledge about health risks and the health system [99]. This first necessary level is reflected in our PLS aims categories Accessibility, Understanding and Knowledge. The second level in Nutbeam's model is called 'interactive' or 'communicative' health literacy. In addition to the abilities required in the first level, the second level refers to the ability to actively communicate about a health topic and derive meaning from health information as well as have the necessary social skills to discuss it with others [99]. This second level is reflected in our PLS aims category Communication of Research. Nutbeam's third level, 'critical health literacy', refers to the ability to critically appraise health information and consequently, have more control over health decisions [99]. This third level is reflected in the PLS aims category 'Empowerment'. There are further similarities between our PLS aims categories and theoretical models that describe product user experience. For example, Morville's honeycomb of user experience [100] includes product findability, accessibility, usability, usefulness, credibility, desirability and value. This and other

models for user design [e.g., 101] have already been applied to evidence summaries by Rosenbaum [51], who emphasizes the importance of 'understandability' in this context. By taking these theoretical considerations together, we conclude that if a PLS's purpose is to reach a wider audience for scientific findings, it may not only be important to consider whether it is accessible for this audience at all, but also to take into account the question of *why* we might want to reach a wider audience with our scientific findings. Is it because scientists simply want readers to understand what they are doing? Or do they even want to enable them to make certain decisions? Carefully considering such questions and reflecting on the aims PLSs are supposed to achieve, ideally, could lead to even more target-oriented research on PLSs, which ultimately improves the development of practical guidelines.

Taking the ontological approach, we strived to evaluate what constitutes a PLS. We found that the characteristics that are discussed and evaluated in the current literature can be classified into the categories Linguistics Attributes, Formal Attributes, General Content, Presentation of Results, Presentation of Quality of Evidence and Contextual Attributes. Subsequently, we pursued a normative approach to determine what a PLS should look like, by reviewing PLS criteria that specify 'values' for certain characteristics that belong to those categories. PLS criteria were mostly related to the characteristics of the Linguistic Attributes, General Content and Formal Structure categories. Some criteria were used to *define* a PLS: In this case, they were meant to distinguish a PLS from a text that is not defined as a PLS (e.g., a PLS is a summary of a study that does not use jargon, while a scientific abstract is a summary of a study that does use jargon [24]). Other criteria were used as indicators for the *quality* of a PLS: These criteria were supposed to distinguish a good or serviceable PLS from a rather unhelpful PLS (e.g., "writing in plain language is not dumbing down the research" [62, p. 5]). While criteria that we linked to characteristics of the Linguistic Attributes category were similar across articles (e.g., avoid jargon and use short sentences [15, 24, 66, 69]) we observed a considerable diversity in those criteria that we linked to characteristics of the Presentation of Results category (e.g., report statistics if meaningful for the target group and/or report statistics in natural frequencies [23, 50, 83], vs. omit any numbers from the PLS [45, 61]). Other criteria were highly specific based on the evidence they reported on. For example, the Cochrane guideline for PLSs states that quality of evidence should be reported based on the GRADE approach [68]. This is because the scientific publications that Cochrane PLSs refer to—systematic reviews and meta-analyses—are all required to report the quality of evidence based on GRADE. Thus, this criterion makes sense for PLSs of original publications that make use of GRADE, but it is of less significance for other PLSs.

Lastly, taking a measurement-related approach, we asked which outcomes are investigated to evaluate PLSs. Outcomes in empirical studies on PLSs mostly related to aims of the Accessibility, Understanding and Knowledge categories. Considering all investigated outcomes, we found that the effects of PLSs were mostly investigated in terms of user experience (i.e., how accessible or understandable PLSs are perceived to be), and with regard to knowledge gains. Thus, we conclude that the aims of the Accessibility, Understanding and Knowledge categories are well represented in empirical investigations of PLSs. However, the questions of whether a PLS is accessible and whether readers understand and gain knowledge from it are closely intertwined, and a differentiated operationalization of these outcomes appears challenging. Furthermore, although aims of the Communication of Research as well as Improvement of Research categories were often mentioned in theoretical articles, the effectiveness of PLSs regarding these aims was scarcely investigated in empirical studies.

Regarding the characteristics, criteria and outcomes, it is hardly surprising that there are significantly fewer theoretical linking points to be identified than to the aims. These three last components of our framework can be considered as tangible consequences of the (theoretical)

aims that are somehow put into practice: The stipulated aims of a PLS determine its characteristics (what is it?), the standards that are applied (how should it be?), and what will be measured (what can it actually achieve?). In the end, all revert to the elemental aims that ultimately lead to creating a PLS: Characteristics form the mere basis for deriving specific values—the criteria—to reach desired outcomes, which are mere operationalization of these aims. Therefore, we conclude that the theoretical basis of PLSs to a very large extent can be attributed to considerations of its various aims and purposes.

Against the background of the conceptual framework, we investigated the existing empirical evidence on PLSs. We found eight studies that investigated PLS criteria by either comparing two different PLS types or by varying specific criteria within the same PLS type. In these studies, effects were observed on outcomes that can be linked to aims of the Accessibility, Knowledge, Understanding, Communication of Research and Empowerment categories. Here, medium text level and guideline-based PLSs were perceived as more accessible [13], and participants preferred such PLSs that were structured, that provided background information and qualitative statements on results that were accompanied by numbers and symbols as well as an evidence quality rating [56, 98]. Furthermore, PLSs with subheadings increased trust in research findings and improved participants' self-reported ability to make decisions [38]. On other outcomes measuring Accessibility, Understanding or Knowledge, however, no statistically significant differences were observed. Changes in the formal structure of PLSs only had a significant impact on knowledge if findings were uncertain, but not if findings were certain [14].

In the empirical studies that were included in our review, we did not find outcomes that could be linked to our theoretically proposed aim category Improvement of Research. This finding indicates that although this aim was named in various theoretical articles dealing with the subject of PLSs, it does not appear to have been empirically evaluated to date. The criteria we found in the theoretical and empirical articles did resemble all proposed characteristic categories. However, they were highly heterogeneous and only rarely empirically investigated.

We further observed that some criteria were frequently listed in guidelines, whereas to our knowledge, they had not been empirically evaluated. For example, most guidelines state that PLSs should be written without jargon [20, 52, 54, 64, 66, 69–73, 75–78]. However, we found no studies that have experimentally varied the use of jargon specifically in PLSs to investigate whether its use affects how laypersons perceive PLSs. It may be worth considering that a PLS which uses jargon while simultaneously providing explanations for such technical terms by adding a glossary might be easier to read and understand than a PLS in which jargon is replaced by lengthy periphrases.

Furthermore, there are conflicting criteria, for example, some guidelines recommend to omit numbers completely [e.g., 27] while others stress the importance of communicating specific numbers [e.g., 68, 69, 92], such as risk ratios. These conflicting criteria might result from disciplinary differences or different aims and priorities concerning the PLSs (e.g., accuracy vs. plainness). Another important finding is that empirical research on PLSs was mostly conducted using small samples of highly educated participants. Of course, when PLSs are investigated, the required sample characteristics depend on the target audience. Since PLSs are meant to communicate science in a way that laypersons can understand, it appears advisable to investigate an approximately representative sample of the population the PLS is directed at. In contrast, investigating characteristics or the efficacy of PLSs in highly educated samples might be fully justified and even desirable if the specific target group of a PLS are researchers from other domains, practitioners, stakeholders or other groups, whose members can be reasonably expected to be highly educated. If this is, however, not the case (e.g., if one wants to reach the public in general), investigating PLSs in highly educated samples imposes a significant threat

on the external validity of the corresponding study. Researchers and providing agencies, therefore, should clearly define and communicate their specific audience and take this adequately into account when evaluating these PLSs.

## Strengths and limitations of the review

This review has some limitations as well as considerable strengths. The fundamental value of this review, at first, lies in its significant theoretical contribution: the proposed conceptual framework that is based on the exhaustive body of scientific literature on PLSs and that includes four basic approaches to comprehensively describe the research field. This framework can be used to design future studies on the effects of PLSs on defined outcomes as well as help develop meaningful PLS guidelines. Second, this is, to our knowledge, the first attempt to map, summarize and discuss the entire current empirical evidence on PLS research. Although a comprehensive statement on the empirical evidence of specific PLS criteria is not (yet) possible, we were able to identify important questions and challenges that may contribute to future research on PLSs and, thus, to the improvement of science communication as a whole. Finally, our twofold approach, developing a conceptual framework and reviewing the empirical evidence on PLSs, made it possible to integrate the latter into the first, putting single empirical findings on certain criteria into a broader explanatory context.

However, there are two main limitations for this study's evidence contribution. First, there is not yet broad consensus about the term 'plain language summary'. Thus, we cannot fully rule out that there might be literature on lay summary formats that we missed in our search due to the fact that we did not know the respective term that was used, even though we made best attempts to be as thorough as possible in our systematic review. Second, there was high heterogeneity among the study designs and outcome measures in the empirical articles we found. Thus, it would not have been appropriate to perform a meta-analysis.

## Conclusions and future research recommendations

Considerable work has been accomplished to establish lay-friendly summary formats that not only communicate scientific findings to laypeople but also simultaneously aim to retain as much scientific rigor and accuracy as possible. Our conceptual framework delineates four main approaches related to aims, characteristics, criteria and outcomes of PLSs to describe the theoretical and empirical research on PLSs, and reveals how they are intertwined in a meaningful way. It thereby constitutes a fertile ground for theory advancement on science communication tools, for hypothesis formulation and testing in empirical studies on PLSs, and for development of writing guidelines. Our findings suggest that in the theoretical as well as the empirical literature, aims of PLSs are clearly named and correspond considerably across articles. Moreover, a significant number of valuable and useful guidelines on writing PLSs are currently available. Also, several studies have empirically evaluated the different types and formats of PLSs with varying criteria on different outcomes. However, their number is relatively low compared to the availability of theoretical literature meticulously outlining the aims and specific characteristics of PLSs. Consequently, we also observed a lack of empirically evaluated criteria in guidelines. A further major implication of our findings is the need for samples in empirical research on PLSs that are approximately representative for the respective target group. Future studies on PLSs should therefore consider their target audience and recruit their samples accordingly. More (ideally, randomized controlled) studies that investigate the effects of single criteria on specific outcomes are also needed. Only then can fully empirically supported recommendations on how to write PLSs be formulated. Such recommendations may complement current guidelines which—at least partly—still lack empirical foundation, as well

as form the basis for the development of new guidelines which may be directed at a specific target group or deal with PLSs from a certain discipline. Altogether, we believe our review to constitute a suitable starting point both for advancing theory on PLSs and for designing and conducting empirical studies on the subject. It is our sincere hope that, in the end, these efforts serve such aims that are meaningful for individuals as well as for society as a whole.

## Supporting information

**S1 Table. Studies included in this review.**
(PDF)

**S2 Table. PLS guidelines and criteria.**
(PDF)

**S3 Table. Quantitative studies comparing PLSs with PLSs: Criteria, outcomes and results.**
(PDF)

**S1 File. Search terms.**
(PDF)

**S2 File. PRISMA checklist.**
(PDF)

**S3 File. Plain language summary of this review.**
(PDF)

## Acknowledgments

We thank Anna Kuznik for her help with preparing the literature analysis tables, as well as Eva Becker and Michelle Bähr for their help with compiling the references and tables. We would also like to thank Lisa Trierweiler for the English proof of the manuscript.

## Author Contributions

**Conceptualization:** Marlene Stoll, Martin Kerwer, Anita Chasiotis.

**Formal analysis:** Marlene Stoll.

**Investigation:** Marlene Stoll, Martin Kerwer, Anita Chasiotis.

**Project administration:** Anita Chasiotis.

**Visualization:** Marlene Stoll.

**Writing – original draft:** Marlene Stoll, Martin Kerwer, Anita Chasiotis.

**Writing – review & editing:** Marlene Stoll, Martin Kerwer, Klaus Lieb, Anita Chasiotis.

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
