## [Decision Letter · Decision Letter 0]

14 Feb 2022

PONE-D-21-24709Plain Language Summaries: A Systematic Review of Theory, Guidelines and Empirical ResearchPLOS ONE

Dear Dr. Stoll,

Thank you for submitting your manuscript to PLOS ONE. After careful consideration, we feel that it has merit but does not fully meet PLOS ONE’s publication criteria as it currently stands. Therefore, we invite you to submit a revised version of the manuscript that addresses the points raised during the review process. Please submit your revised manuscript by Mar 31 2022 11:59PM. If you will need more time than this to complete your revisions, please reply to this message or contact the journal office at plosone@plos.org. Please include the following items when submitting your revised manuscript:A rebuttal letter that responds to each point raised by the academic editor and reviewer(s). You should upload this letter as a separate file labeled 'Response to Reviewers'.A marked-up copy of your manuscript that highlights changes made to the original version. You should upload this as a separate file labeled 'Revised Manuscript with Track Changes'.An unmarked version of your revised paper without tracked changes. You should upload this as a separate file labeled 'Manuscript'.

We look forward to receiving your revised manuscript.

Kind regards,

Kelvin Ian Afrashtehfar, M.Sc., D.D.S.,Dr. med. dent., FRCDC

Academic Editor

PLOS ONE

Journal Requirements:

5. Your abstract cannot contain citations. Please only include citations in the body text of the manuscript, and ensure that they remain in ascending numerical order on first mention.

Additional Editor Comments (if provided):

Dear Editors,

A further revision is needed after appraising the manuscript in detail and obtaining feedback from the reviewers.

Academic Editor

Reviewers' comments:

Reviewer's Responses to Questions

**Comments to the Author**

1. Is the manuscript technically sound, and do the data support the conclusions?

Reviewer #1: Partly

Reviewer #2: Yes

Reviewer #3: Yes

2. Has the statistical analysis been performed appropriately and rigorously? 

Reviewer #1: N/A

Reviewer #2: N/A

Reviewer #3: N/A

3. Have the authors made all data underlying the findings in their manuscript fully available?

Reviewer #1: Yes

Reviewer #2: Yes

Reviewer #3: Yes

4. Is the manuscript presented in an intelligible fashion and written in standard English?

Reviewer #1: Yes

Reviewer #2: Yes

Reviewer #3: No

5. Review Comments to the Author

Reviewer #1: The authors provide an overview of published and available records reporting on PLS.

Overall, the background and the purpose of this systematic review is well explained. However, I find some overlap between the methods and the results sections with very long sentences, that are difficult to follow. I would also simplify the Objectives sections and try to synthesise it further to make it more digestible.

Did the authors perform a quality control on the included records?

As mentioned previously, I think the text can be summarised more:

For example, lines 306-308 are part of the methods, should start directly with “PLS aims can be divided...”

•The same for lines 365-367, should start directly with “PLS characteristics can be divided...”

•The same for Criteria lines 414-419, this info should be part of the methods and not the results

•The same for Outcomes lines 484-489

It is unclear what Criteria of PLS represent and how it differs from Characteristics, there seems to be something between Characteristics and quality of the information provided. A more detailed explanation should be provided.

The authors mentioned (line 424-425) that they matched the criteria extracted from the articles to the respectively fitting characteristic category. it's unclear how they have done it, not really explained in the methods. Also, reporting a table of these findings would help to understand how characteristics and criteria are related.

Similar to my previous comment, a descriptive table of how characteristics, criteria, aims and outcomes are related would make the findings more understandable instead of having everything reported in text.

S 1 Table should provide more details as reported in the manuscript and distinguish between “Empirical that evaluated one type of PLS”, “Empirical articles that compared one type of PLS to other summary formats” and “Empirical Articles that compared different forms of PLS”.

Reviewer #2: The authors have done a very good work with this systematic review as it is explicit and explains the subject matter in a clear and logical manner. The study describes what has previously been done in an easy to understand way and summaries the references very well.

Reviewer #3: my dear author ;

strength:

Many thanks for this great work that's is well written and you do a big effort to map the first attempt to discuss the empirical evidence on PLS, and a good framework was done which built the basis for future studies and the systematic review was included a suitable number of articles (90 records), double screening of study selection, so this article has many strong points like simplicity, feasibility, the brainpower

weak points :

1- references have No DOI which remembered at instructions of the journal

2- I wish the author if do meta-analysis and statistics to complete this great work and the article will be a systematic review and metanalysis

3- English language and grammar need some correction

4- no risk of bias among studies and individual is available at Prisma checklist

finally many thanks for good efforts and great work for summarising and planning for PLS framework for future guidelines and studies

6. PLOS authors have the option to publish the peer review history of their article (what does this mean?). If published, this will include your full peer review and any attached files.

Reviewer #1: No

Reviewer #2: No

Reviewer #3: **Yes: **Amr Kamel Khalil Ahmed

---

## [Author Response · Author response to Decision Letter 0]

30 Mar 2022

Dear editor, dear reviewers of our manuscript, 

We thank you for your helpful and insightful comments. We revised our manuscript accordingly and do now submit a new version which we believe has benefited greatly from your suggestions. 

Please find below our responses to your comments. We marked each of your comments with “Rx” (or "Ex") and our responses with “RRx” (or "REx"). Line numbers refer to the PDF Proof version. 

Editor Comments

E1. Please ensure that your manuscript meets PLOS ONE's style requirements, including those for file naming. The PLOS ONE style templates can be found at https://journals.plos.org/plosone/s/file?id=wjVg/PLOSOne_formatting_sample_main_body.pdf and https://journals.plos.org/plosone/s/file?id=ba62/PLOSOne_formatting_sample_title_authors_affiliations.pdf

RE1: Thank you for this advice. We corrected the manuscript title and deleted the subheading in the abstract as well as in the subsections “Results - Aims of PLS”, “Results - Characteristics of PLS” and “Results - Empirical articles that compared different forms of PLS”. We also revised the reference list and corrected some minor deviations from the reference style.

E2. Please update your submission to use the PLOS LaTeX template. The template and more information on our requirements for LaTeX submissions can be found at http://journals.plos.org/plosone/s/latex.

RE2: We did not submit a LaTex manuscript. 

E3. We note that the grant information you provided in the ‘Funding Information’ and ‘Financial Disclosure’ sections do not match. When you resubmit, please ensure that you provide the correct grant numbers for the awards you received for your study in the ‘Funding Information’ section.

RE3: We did not receive any grants or third-party funding for this study. We revised the “Funding Information” and “Financial Disclosure” sections accordingly. 

E4. In your Data Availability statement, you have not specified where the minimal data set underlying the results described in your manuscript can be found. PLOS defines a study's minimal data set as the underlying data used to reach the conclusions drawn in the manuscript and any additional data required to replicate the reported study findings in their entirety. All PLOS journals require that the minimal data set be made fully available. For more information about our data policy, please see http://journals.plos.org/plosone/s/data-availability.

RE4: As we conducted a systematic literature review in a narrative way and this did not involve acquisition of any empirical data or any formal statistical analysis, there are no data we could provide and upload to a data repository. The results described in our manuscript are summaries of the studies listed in S1 Table, and the search queries we used to find these studies is available in S1 File (“All relevant data are within the manuscript and its Supporting Information files.”)

E5. Your abstract cannot contain citations. Please only include citations in the body text of the manuscript, and ensure that they remain in ascending numerical order on first mention.

RE5: Thank you for pointing this out. There were no citations in our abstract, but we numbered our research objectives with “(1)” and “(2)”. We revised this sentence so that now no numbers in brackets appear in the abstract: 

“The two main objectives of this review were to develop a conceptual framework for PLS theory, and to synthesize empirical evidence on PLS criteria.” (l. 23 ff.)

Reviewer Comments

Reviewer 1: 

The authors provide an overview of published and available records reporting on PLS.

R1.1a: Overall, the background and the purpose of this systematic review is well explained. However, I find some overlap between the methods and the results sections with very long sentences, that are difficult to follow. 

R1.1b: I would also simplify the Objectives sections and try to synthesise it further to make it more digestible.

RR1.1 Thank you for pointing this out. We revised the whole manuscript accordingly and shortened the Objectives section and parts of the Methods section. To achieve a clearer separation between the Introduction, Methods and Results section, we now only explain the four main subject areas (aims, characteristics, criteria and outcomes) in the Introduction section (l. 133 ff.; l. 170 ff.) and summarize all information on our methodological approach in the Methods section (l. 245 ff.). The Results section is now also more streamlined, since we added two tables (Table 1, Table 2) to structure the findings and rearranged the subsections. Also, we submitted our manuscript to a native speaker language editor for proofreading and revised it accordingly, hoping that this will make the manuscript more digestible.

R1.2: Did the authors perform a quality control on the included records?

RR1.2: Our aim was to give a descriptive overview of all existing literature on PLS and to describe the current state of research. Therefore, we did not perform an additional quality assessment of the included reports. We added an explanatory sentence in the Methods section:

“As our aim was to give a descriptive overview of the literature on PLSs, we did not perform an additional quality assessment of the reports.” (l. 237-239)

R1.3a: As mentioned previously, I think the text can be summarised more:

For example, lines 306-308 are part of the methods, should start directly with “PLS aims can be divided...”

R1.3b: The same for lines 365-367, should start directly with “PLS characteristics can be divided...”

R1.3c: The same for Criteria lines 414-419, this info should be part of the methods and not the results

R1.3d: The same for Outcomes lines 484-489

RR1.3: Thank you for these suggestions. We moved this information from the Results section to the revised Methods section (l. 254 ff.). 

R1.4: It is unclear what Criteria of PLS represent and how it differs from Characteristics, there seems to be something between Characteristics and quality of the information provided. A more detailed explanation should be provided.

RR1.4: Thank you for this suggestion. In order to make the distinction between characteristics and criteria more clear, we explained our methodological approach in more detail (see also RR1.5). 

“We discovered that outcomes constitute operationalizations of aims and were therefore subordinated to the aims categories. Similarly, criteria appeared to be operationalizations and specifications of characteristics. In the next step, the three experienced authors proposed categories for the subject areas ‘aims’ and ‘characteristics’ only.“ (l. 257-261) 

We further added tables and rearranged the subsections in the Results section. Additionally, we discussed this issue with the native speaker language editor at our institute. To clarify how we understand the terms that we used in the manuscript, we added an analogy :

“We illustrate the specific meaning of these terms within this article with the following analogy: Assume the topic is not PLS but a cake you want to bake for guests. A cake’s aim or purpose could be to “taste delicious”. Characteristics of a cake are, among others, its ingredients (“flour”, “sugar”). Criteria are target values of the characteristics that aim to fulfill its purpose (“3 cups flour”, “2 cups sugar”). An outcome to test whether the “aim” of the cake is achieved is to ask your guests if the cake tasted delicious.” (l. 137-143)

R1.5: The authors mentioned (line 424-425) that they matched the criteria extracted from the articles to the respectively fitting characteristic category. it's unclear how they have done it, not really explained in the methods. Also, reporting a table of these findings would help to understand how characteristics and criteria are related.

RR1.5. Thank you for this helpful feedback. In order to clarify our approach, we revised the Methods / Analysis section (l. 245 - 276). We now describe the process of matching criteria to characteristic categories and outcomes to aim categories in more detail (“We also mapped the information on criteria to the characteristic categories of PLSs to determine the specifications of different categories of PLS characteristics (i.e., criteria) that have been found to or are suggested to distinguish a PLS from other text formats or to constitute a high-quality PLS.” l. 265-268). Therefore, we added two new tables with prototypical examples of text passages and the respective category for aims and outcomes (Table 1) as well as for characteristics and criteria (Table 2). With this, we hope to make the relationship and differences between aims and outcomes as well as between characteristics and criteria more clear.

R1.6: Similar to my previous comment, a descriptive table of how characteristics, criteria, aims and outcomes are related would make the findings more understandable instead of having everything reported in text.

RR1.6: We hope that our efforts with the new tables (Table 1, Table 2) helped to make our findings more understandable (see RR1.5). 

R1.7: S 1 Table should provide more details as reported in the manuscript and distinguish between “Empirical that evaluated one type of PLS”, “Empirical articles that compared one type of PLS to other summary formats” and “Empirical Articles that compared different forms of PLS”.

RR1.7: Thank you for this suggestion. We added these details in the S1 Table.

Reviewer 2:

R2.1: The authors have done a very good work with this systematic review as it is explicit and explains the subject matter in a clear and logical manner. The study describes what has previously been done in an easy to understand way and summaries the references very well.

RR2.1: We thank the reviewer for this encouraging comment.

Reviewer 3:

my dear author ;

strength:

Many thanks for this great work that's is well written and you do a big effort to map the first attempt to discuss the empirical evidence on PLS, and a good framework was done which built the basis for future studies and the systematic review was included a suitable number of articles (90 records), double screening of study selection, so this article has many strong points like simplicity, feasibility, the brainpower

weak points :

R3.1: references have No DOI which remembered at instructions of the journal

RR3.1: Thank you for pointing this out. We added all available DOI numbers in the reference list.

R3.2: - I wish the author if do meta-analysis and statistics to complete this great work and the article will be a systematic review and metanalysis

RR3.2: We appreciate this comment since at first, our goal was indeed to conduct a meta-analysis complementing the systematic review. Unfortunately, the empirical studies we found were too heterogeneous and diverse with respect to their design to be combined into a meta-analysis. To make this more clear to our readers, we added a statement in the Results section: 

“The included empirical articles on PLS were highly heterogeneous with respect to study design, use of terminology and operationalization. Therefore, it was not possible to conduct a meta-analysis to quantitatively synthesize the study effects.” (l. 527 - 534)

And in addition to this paragraph in the Discussion section: 

“Second, there was high heterogeneity among the study designs and outcome measures in the empirical articles we found. Thus, it would not have been appropriate to perform a meta-analysis.” (l. 862 - 864))

R3.3: - English language and grammar need some correction

RR3.3: Thank you for this advice. We submitted our manuscript to a native speaker language editor for proofreading and revised it accordingly.

R3.4:- no risk of bias among studies and individual is available at Prisma checklist

RR3.4: This is because we did not perform a risk of bias assessment. To make this more clear to our readers, we added the following sentence in the Methods section: 

“As our aim was to give a descriptive overview of the literature on PLSs, we did not perform an additional quality assessment of the reports.” (l. 237 - 239)

---

## [Decision Letter · Decision Letter 1]

9 May 2022

Plain language summaries: a systematic review of theory, guidelines and empirical research

PONE-D-21-24709R1

Dear Dr. Stoll,

We’re pleased to inform you that your manuscript has been judged scientifically suitable for publication and will be formally accepted for publication once it meets all outstanding technical requirements.

Kind regards,

Kelvin I. Afrashtehfar, M.Sc., D.D.S.,Dr. med. dent., FRCDC

Academic Editor

PLOS ONE

Additional Editor Comments (optional):

Dear Authors,

I have just received the last reviewer’s comments.

After thoroughly appraising your manuscript, we have concluded that your work is suitable for publication. Therefore, it has been granted an acceptance.

Should you dispute this decision, do not hesitate to contact me for further elaboration.

Kind Regards,

The Academic Editor

Afrashtehfar KI

Reviewers' comments:

Reviewer's Responses to Questions

**Comments to the Author**

1. If the authors have adequately addressed your comments raised in a previous round of review and you feel that this manuscript is now acceptable for publication, you may indicate that here to bypass the “Comments to the Author” section, enter your conflict of interest statement in the “Confidential to Editor” section, and submit your "Accept" recommendation.

Reviewer #1: All comments have been addressed

Reviewer #2: All comments have been addressed

Reviewer #3: All comments have been addressed

2. Is the manuscript technically sound, and do the data support the conclusions?

Reviewer #1: Yes

Reviewer #2: Yes

Reviewer #3: Yes

3. Has the statistical analysis been performed appropriately and rigorously? 

Reviewer #1: Yes

Reviewer #2: N/A

Reviewer #3: Yes

4. Have the authors made all data underlying the findings in their manuscript fully available?

Reviewer #1: Yes

Reviewer #2: Yes

Reviewer #3: Yes

5. Is the manuscript presented in an intelligible fashion and written in standard English?

Reviewer #1: Yes

Reviewer #2: Yes

Reviewer #3: Yes

6. Review Comments to the Author

Reviewer #1: The authors have address my suggestions and feedback in this revised version. I have no further comments.

Reviewer #2: The subject matter is highly relevant and will be beneficial to a wide range of researchers and readers. Also, the authors have adequately addressed all comments that were raised during the review of the manuscript.

Reviewer #3: many thanks to the authors and for your nice responses, you gave convincing responses, and wish future articles for you

7. PLOS authors have the option to publish the peer review history of their article (what does this mean?). If published, this will include your full peer review and any attached files.

Reviewer #1: No

Reviewer #2: No

Reviewer #3: **Yes: **Amr Kamel Khalil Ahmed

---

## [Editor Report · Acceptance letter]

27 May 2022

PONE-D-21-24709R1 

Plain language summaries: a systematic review of theory, guidelines and empirical research 

Dear Dr. Stoll:

I'm pleased to inform you that your manuscript has been deemed suitable for publication in PLOS ONE. Congratulations! Your manuscript is now with our production department. 

Kind regards, 

on behalf of

Dr. Kelvin I. Afrashtehfar 

Academic Editor

PLOS ONE